# An ethylene biosynthesis enzyme controls quantitative variation in maize ear length and kernel yield

Qiang Ning[1,3], Yinan Jian[1,3], Yanfang Du[1], Yunfu Li[1], Xiaomeng Shen[1], Haitao Jia[1], Ran Zhao[1], Jimin Zhan[1], Fang Yang[1], David Jackson [1,2✉], Lei Liu [1,2✉] & Zuxin Zhang [1✉]

Maize ear size and kernel number differ among lines, however, little is known about the molecular basis of ear length and its impact on kernel number. Here, we characterize a quantitative trait locus, *qEL7*, to identify a maize gene controlling ear length, flower number and fertility. *qEL7* encodes 1-aminocyclopropane-1- carboxylate oxidase2 (ACO2), a gene that functions in the final step of ethylene biosynthesis and is expressed in specific domains in developing inflorescences. Confirmation of *qEL7* by gene editing of *ZmACO2* leads to a reduction in ethylene production in developing ears, and promotes meristem and flower development, resulting in a ~13.4% increase in grain yield per ear in hybrids lines. Our findings suggest that ethylene serves as a key signal in inflorescence development, affecting spikelet number, floral fertility, ear length and kernel number, and also provide a tool to improve grain productivity by optimizing ethylene levels in maize or in other cereals.

[1] National Key Laboratory of Crop Genetic Improvement, Hubei Hongshan Laboratory, Huazhong Agricultural University, Wuhan, People's Republic of China. [2] Cold Spring Harbor Laboratory, Cold Spring Harbor, New York, NY, USA. [3]These authors contributed equally: Qiang Ning, Yinan Jian. ✉email: jacksond@cshl.edu; lliu@cshl.edu; zuxinzhang@mail.hzau.edu.cn

Maize (*Zea mays* L.) is the world's most productive cereal crop, partly due to the development of large ears with hundreds of kernels which develop in a stereotypical pattern from fertile florets. The mature female inflorescences of maize usually bear hundreds of kernels, which provide substantial nutrition to meet increasing demands for dietary energy and biofuel consumption by an increasing human population[1,2]. Promoting maize grain yield per unit area by genetic approaches is a sustainable path to double the global supply of maize grains. For high-yield maize breeding, kernel number per ear is one of the key breeding targets[3]. Kernels arise from well-developed and pollinated female florets, borne from spikelet meristems (SMs) derived from the inflorescence meristem (IM)[4,5]. Therefore, IM activity is believed to determine the number of florets and kernels on the maize inflorescence.

A complex functional hierarchy of genes participate in IM maintenance, including the classical *CLAVATA-WUSCHEL* (*CLV-WUS*) feedback signaling pathway[6]. Phytohormone signaling is also involved in regulating the activity of IMs in maize. Auxin biosynthesis-related genes *SPARSE INFLORESCENCE 1* (*SPI1*)[7] and *VANISHING TASSEL2* (*VT2*)[8], and auxin signaling genes *BARREN INFLORESCENCE 1* (*BIF1*), *BIF2* and *BIF4*[9,10] influence the initiation of the spikelet paired meristems (SPMs), and mutants in these genes reduce the number of florets on the ear. In addition to auxin, ethylene also regulates many aspects of plant growth and development, including flower development and sex determination[11–15]. In maize, the pleiotropic effect of ethylene was discovered by spraying plants with an appropriate concentration of Ethephon, which is converted to ethylene and is effective as an anti-lodging agent for small grains. Ethephon can enhance the grain yield when lodging is a problem[16]. In the absence of lodging, low rates of Ethephon can also increase grain yield, but high rates of Ethephon decrease yield[16,17]. Repression of ethylene biosynthesis or response by genetic manipulation enhances maize grain yield under drought and low nitrogen conditions[18–20], indicating that maize grain yield can be enhanced by modifying ethylene synthesis and signaling. However, the ethylene-related genes that influence grain yield remain largely unknown.

## Results and discussion

### *qEL7* controls grain yield by affecting floret number and fertility in developing ears.
Previously, we identified a major QTL, *qEL7*, for maize ear length and kernel number and developed a set of QTL near-isogenic lines (NILs) (*qEL7^SL17^* and *qEL7^Ye478^*)[21]. *qEL7^Ye478^* plants had a significant increase in ear length (EL), kernel number per row (KNR), 100-kernel weight and ear weight (EW), and later flowering, compared with *qEL7^SL17^* (Fig. 1a−g and Supplementary Table 1), although they were similar in kernel row number, ear diameter, plant height, leaf length and width (Fig. 1d, e, Supplementary Fig. 1a and Supplementary Table 1). Despite having longer ears with more seeds at maturity, developing *qEL7^Ye478^* ears had significantly shorter ear IMs and fewer florets (Fig. 1h, j, k, Supplementary Fig. 1b and Supplementary Table 1). Maize ears are indeterminate, producing florets at the tip that are not fertilized or "filled" to produce seed, and these unproductive or aborted florets are often masculinized. By careful analysis of developing ears, we found that the floret abortion rate of the florets in the *qEL7^Ye478^* long ear NILs (~18.7%) was dramatically lower than in the *qEL7^SL17^* short ear NILs (~44.7%) (Fig. 1i, l, m, Supplementary Fig. 1c and Supplementary Table 1). These results indicate that additional kernels in the long ear *qEL7^Ye478^* result from a higher proportion of florets that developed into seeds, not from increased floret production.

### SNPs and InDels in the *ZmACO2* promoter are candidate variants underlying *qEL7*.
To fine map *qEL7*, we identified seven close recombinants from mapping populations derived from the QTL parents and conducted progeny testing (Supplementary Fig. 1d). By fine mapping, we delineated *qEL7* into a 50.8-kb physical interval flanked by markers M4 and M5 (Fig. 2a, Supplementary Fig. 1d and Supplementary Data 1). Homozygous recombinants carrying the *qEL7^Ye478^* allele in this interval showed an increase in EL, KNR, and EW, but no change in kernel row number across two environments (Supplementary Fig. 1d and Supplementary Data 1). Only one gene, *Zm00001d020686*, which encodes 1-aminocyclopropane-1-carboxylate oxidase2 (ACO2), is annotated in this region of B73 RefGenV4 (Fig. 2a), and was referred to as *ZmACO2*. Sequencing of *ZmACO2* from the QTL parents revealed five insertion/deletions (InDels) and eleven single nucleotide polymorphisms (SNPs) in its 5′-untranslated region (UTR) and promoter region, but complete sequence identity in the coding region (Fig. 2b). Consistent with these differences in non-coding regions, expression of *ZmACO2* in developing ears differed in the parental lines, with lower expression in the long ear *qEL7^Ye478^* line than in the short ear *qEL7^SL17^* line ($P < 0.01$) (Fig. 2c). These results suggest that sequence variation in the promoter leads to the differences in *ZmACO2* expression and ear phenotypes between *qEL7^Ye478^* and *qEL7^SL17^*.

We next asked if sequence variations of *ZmACO2* were associated with ear-traits. We sequenced the 3,539 bp genomic region, covering the promoter and gene body of *ZmACO2*, in 214 diverse inbred lines. We identified 56 polymorphic variants (Fig. 2d), and found that four SNPs (-666, -645, -643 and -622) and one 7 bp InDel (-298) in the promoter region were significantly associated with ear length ($p = 1.73E−04$) (Supplementary Table 2). Transcription factor (TF) binding motif predictions showed that the 7 bp insertion could introduce new binding sites for bHLH, TCP, and Dehydrin TF families (Supplementary Table 3). The five associated sites, which were in complete linkage disequilibrium, formed two haplotypes: one with *qEL7^SL17^* genotype and the other with *qEL7^Ye478^* genotype were named *Hap^SL17^* and *Hap^Ye478^*, respectively (Fig. 2d). Those inbred lines carrying *Hap^Ye478^* exhibited a longer ear with more kernels per row than inbred lines carrying *Hap^SL17^* (Fig. 2e, f), and expression of *ZmACO2* in lines with *Hap^Ye478^* was significantly lower than in the *Hap^SL17^* lines (Fig. 2g). Therefore, we infer that the five associated variants in the *ZmACO2* promoter are possible causal sites that contribute to variations of *ZmACO2* expression and the ear-trait phenotypes. We next sequenced the *ZmACO2* promoter and gene body from 53 teosinte and 44 maize landrace lines to examine the selection pressure acting on this region during maize domestication and improvement (Supplementary Data 3). The promoter region, covering the five associated sites, showed a weak signal of selection, with a considerable reduction in nucleotide diversity ($\pi_{maize}/\pi_{teosinte} = 0.33$) and a non-neutral evolution pattern ($p = 0.02$, HKA test) from teosinte to maize (Supplementary Fig. 2a). However, the favorable *Hap^Ye478^* haplotype wasn't obviously enriched from teosinte to maize, as it was present in over 50% of both teosinte and maize lines (Supplementary Fig. 2b), indicating that *Hap^Ye478^* emerged before domestication and was maintained at a high frequency during maize domestication and improvement.

### *ZmACO2* negatively controls ear length and grain yield.
To confirm the function of *ZmACO2* in ear development, we created two knockout lines (*aco2-cr1* and *cr2*) by CRISPR-Cas9 (Fig. 3a−d and Supplementary Fig. 3a) and three overexpression

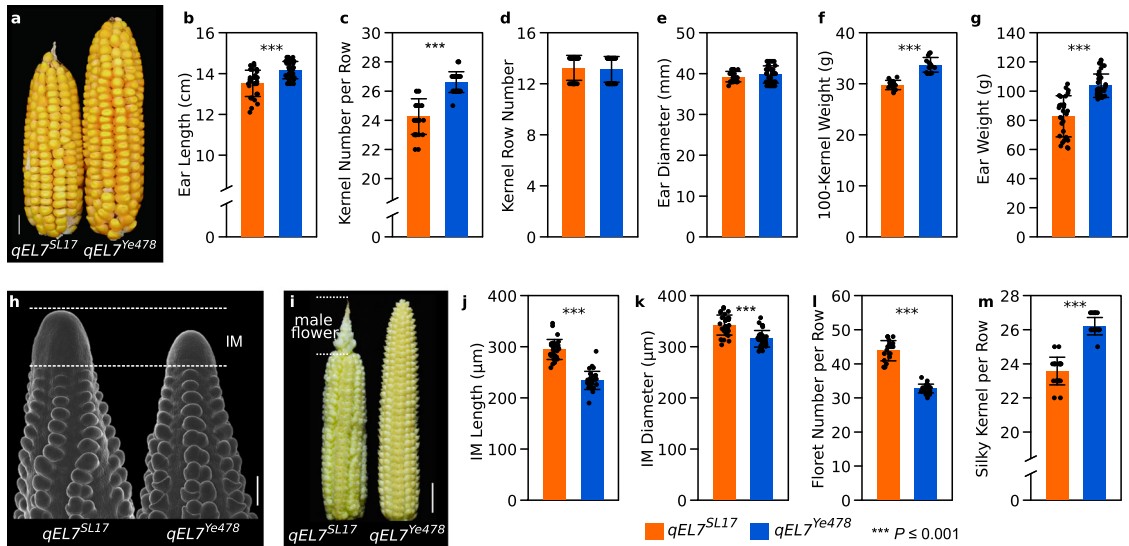

**Fig. 1 qEL7 moderates grain yield by affecting the proportion of florets that developed into seeds during ear development.** $qEL7^{Ye478}$ plants make longer ear with more kernels per row (**a**, scale bar = 1 cm), smaller IM (**h**, scale bar = 200 μm), and higher floret fertility (**i**, scare bar = 2 cm) than that in $qEL7^{SL17}$. Phenotypes of ear length (**b**, $p = 1.82 \times 10^{-6}$), kernel number per row (**c**, $p = 1.03 \times 10^{-15}$), kernel row number (**d**, $p = 0.30$), ear diameter (**e**, $p = 0.10$), 100-kernel weight (**f**, $p = 7.42 \times 10^{-10}$) and ear weight (**g**, $p = 1.90 \times 10^{-11}$) in $qEL7^{SL17}$ and $qEL7^{Ye478}$; $n = 39$ and 32, respectively. IM length (**j**, $p = 5.92 \times 10^{-19}$) and diameter (**k**, $p = 1.04 \times 10^{-7}$) of $qEL7^{SL17}$ and $qEL7^{Ye478}$; $n = 31$ for both. Floret number per row (**l**, $p = 1.24 \times 10^{-20}$) and silky kernels per row (**m**, $p = 9.04 \times 10^{-17}$) of the immature ears before pollination in $qEL7^{SL17}$ and $qEL7^{Ye478}$; $n = 21$ and 24, respectively. For **b**−**g** and **j**−**m** data are presented as means ± SD. ***p-value ≤ 0.001, from a two-tailed, two-sample t-test; orange bar, $qEL7^{SL17}$; blue bar, $qEL7^{Ye478}$. For **h**, ears from five plants of each genotype were imaged by scanning electron microscope. $n$ is the number of ears examined in (**b**−**g**) and (**j**−**m**).

lines (ACO2-OE1, OE2, and OE3) (Fig. 3e−h) using maize Ubiquitin promoter-driven ZmACO2 constructs (pUBI:ZmACO2). ZmACO2 loss-of-function lines showed an increase in ear length by 6.8−7.2% (Fig. 3b), in kernel number per row by 7.9−15.2% (Fig. 3c), and in ear weight by 13.9−15.1% (Fig. 3d), compared with the corresponding wild-type siblings. Through the analysis of developing ears, we found aco2-cr1 plants had a lower floret abortion rate (Supplementary Fig. 3b−g), consistent with the effect of the long ear allele $qEL7^{Ye478}$ in floret fertility (Supplementary Fig. 1c). However, the aco2-cr1 plants also had a larger IM with more florets. These results suggest that aco2 knockout alleles increase kernel number by promoting both floret formation and floret fertility. The aco2 knockout lines also did not impact other leaf phenotypes, such as senescence (Supplementary Fig. 3h). Overexpressing ZmACO2 led to a 3−5 fold elevation of transcript levels (Supplementary Fig. 3i), and a decrease in ear length by 9.3−10.9% (Fig. 3f), in kernel number per row by 8.8−16.0% (Fig. 3g), and in ear weight by 21.7−25.2% (Fig. 3h) relative to the wild-type siblings. We confirmed the ear length and kernel number per row changes of overexpression and knockout lines in a second field season (Supplementary Data 4). These results confirmed that ZmACO2 negatively controls kernel number and grain yield in maize.

In addition to knockout and overexpression of ZmACO2, we also mutated its promoter by CRISPR-Cas9 using eight single-guide RNAs (sgRNAs) to target a 520 bp non-repetitive promoter region. The targeted region covered the candidate causal 7 bp InDel (-298) of qEL7, and accessible chromatin detected in developing ears by Assay for Transposase-Accessible Chromatin using sequencing[22] and Micrococcal Nuclease digestion with deep sequencing[23] (Supplementary Fig. 3j). We identified five promoter-edited alleles ($ACO2^{CR-pro1}$ to $ACO2^{CR-pro5}$) with 71−516 bp deletions in this region (Supplementary Fig. 3k) that decreased ZmACO2 expression in developing ears (Supplementary Fig. 3l). Like aco2-cr knockout alleles, these promoter-edited alleles showed a significant increase in ear length by 6.1−14.7%,

in kernel number per row by 8.6−17.8% and in ear weight by 14.6−24.7%, compared to wild-type siblings (Supplementary Fig. 3m−o and Supplementary Data 5). Similar changes of these promoter-edited lines in ear length and kernel number per row were observed in a second field season (Supplementary Data 5). Therefore, this promoter region is critical for maintaining ZmACO2 expression, and its disruption increased ear length and kernel number.

**ZmACO2 regulates ethylene biosynthesis in developing ears.** ZmACO2 was expressed widely in roots, internodes, and leaves, with a high expression in inflorescences (Supplementary Fig. 4a). RNA in situ hybridization revealed that ZmACO2 transcripts were enriched in the adaxial domains of SPMs (Fig. 4a), in semicircular domains at the base of SMs (Fig. 4b), and at the junction between glumes and FMs (Fig. 4c), suggesting a highly localized expression pattern is important for its developmental function. We next investigated the molecular function of ZmACO2. ACC oxidase converts ACC (1-aminocyclopropane-1-carboxylate) into ethylene in the final pathway step of ethylene biosynthesis in plants (Fig. 4e)[24]. To confirm the biological activity of ZmACO2, we purified ZmACO2 protein and measured its enzyme activity in vitro by quantifying ethylene production at different ACC substrate concentrations. A conserved histidine participates in the binding of a $Fe^{2+}$ cation required for ACO catalytic activity[25], so we made a negative control ZmACO2 protein with a histidine (H) to glutamine (Q) substitution at this residue ($H_{186}Q$) (Supplementary Fig. 4b). The amount of ethylene production catalyzed by ZmACO2 increased exponentially up to an ACC concentration of 20 mM, whereas the negative control $H_{186}Q$ proteins had no activity, as expected (Fig. 4f). Next, to ask if ZmACO2 controlled ethylene biosynthesis in vivo, we measured ethylene levels in dissected developing ears, and found that ethylene levels were significantly lower in the long ear $qEL7^{Ye478}$ line than that in $qEL7^{SL17}$ (Fig. 4g), which agreed with ZmACO2 expression in the two NILs (Fig. 2c). We further observed

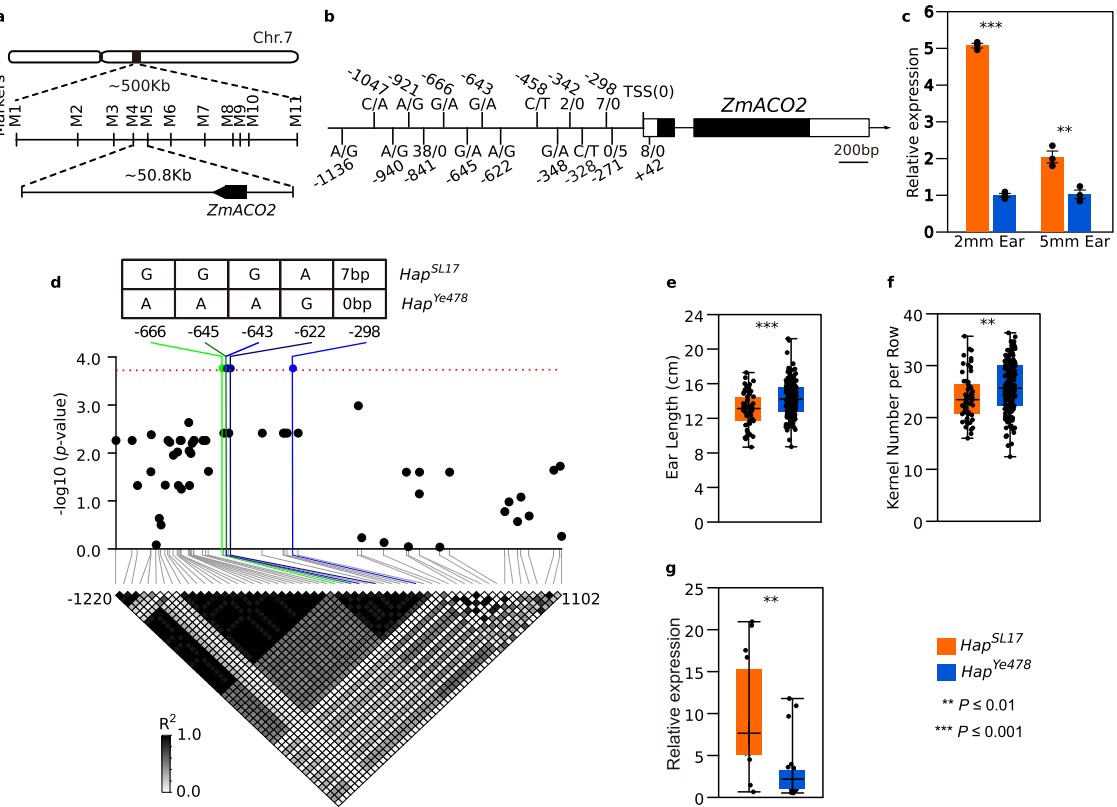

**Fig. 2 ZmACO2 is the candidate gene of qEL7 and negatively affects yield-related traits. a** By fine mapping, qEL7 was delineated into a 50.8 kb interval with only one predicted gene ZmACO2; detailed information is presented in Supplementary Fig. 1d. **b** Sequence variations identified in ZmACO2 genomic regions between NILs; the positions (relative to transcript start site, TSS) were listed beside genotypes information; the genotypes were presented as qEL7SL17/qEL7Ye478; the nucleotide bases and numbers represent the genotypes of SNPs and base pairs of InDels, respectively; white box, UTR region; black box, exon; arrowhead, gene direction. **c** ZmACO2 expression was lower in the 2 mm ($p = 4.33 \times 10^{-7}$) and 5 mm ($p = 0.0034$) ears of qEL7Ye478 (blue bar) than that in qEL7SL17 (orange bar), data are presented as means ± SD and **$p$-value ≤ 0.01, ***$p$-value ≤ 0.001, from a two-tailed, two-sample t-test. **d** Five variants in ZmACO2 promoter show association with ear length in diversity inbred lines; the red dash line indicates the threshold of significant association at $p ≤ 1.79 \times 10^{-4}$; the white to black heatmap shows the pairwise linkage disequilibrium pattern by $R^2$; the genotypes and positions (relative to TSS) of five significant association loci are shown on the top and indicated by green and blue dash lines. HapYe478 haplotype has a longer ear (**e**, $p = 6.94 \times 10^{-5}$), more kernel number per row (**f**, $p = 0.0041$) and lower ZmACO2 expression (**g**, $p = 0.0014$, H = 10.26) than HapSL17 haplotype among diverse inbred lines; orange box, HapSL17 haplotype ($n = 56$ in **e** and **f**, $n = 16$ in **g**); blue box, HapYe478 haplotype ($n = 158$ in **e** and **f**, $n = 24$ in **g**); each box represents the median and interquartile range, and whiskers extend to maximum and minimum values; **$p$-value ≤ 0.01, ***$p$-value ≤ 0.001 from a two-tailed, two-sample t-test (**e**, **f**) and Kruskal–Wallis test (**g**). For expression level is measured by qRT-PCR; three biological replicates and three technical replicates in (**c**); one biological replicate and three technical replicates for each line in (**g**); approximately 20 ears were used in each biological replicate. $n$ is the number of inbred lines examined in (**e**–**g**).

decreased ethylene levels in the aco2-cr1 line, and increased ethylene levels in the ACO2-OE3 line (Fig. 4h), supporting the idea that ZmACO2 catalyzes ACC conversion into ethylene in vivo. Together, these results suggest that ZmACO2 negatively controls kernel number and yield in maize by regulating ethylene biosynthesis. Phytohormones play crucial roles in plant growth, development, and response to various stimuli. The gaseous phytohormone ethylene has long been recognized as a growth inhibitor[26,27]. However, ethylene was found to promote cell division in the quiescent center in the root stem cell niche in Arabidopsis and rice[28,29]. Ethylene also has a dose-dependent effect in opposite directions, both promoting and repressing cell division and fruit elongation in cucumber[15]. Therefore, ethylene appears to have alternating roles, either stimulating or inhibiting growth, depending on endogenous and environmental conditions and developmental stage[26,27]. Here, a higher ethylene level could increase IM size and form more spikelets in developing qEL7SL17 ears compared to qEL7Ye478. Usually, IM size is positively correlated with kernel numbers and grain yield in maize[30–33]. However, sex determination of florets was abnormal in qEL7SL17

ears, and associated with a reduction in floret fertility, ear length, kernel number, and grain yield, suggesting that floret fertility is also critical for maize grain yield. These findings suggest the role of ethylene in maize grain yield-related traits is more complex than in cucumber fruit size regulation[15], and the optimization of ethylene levels for grain yield enhancement of maize or other cereal crops needs to balance its effect in both meristem activity and floret fertility.

**Ethylene may participate in cross-talk with other phytohormones in developing ears.** Recent findings suggest that the function of ethylene in primary root elongation is mediated via interactions with other phytohormones, such as auxin, gibberellin (GA), cytokinins (CK), jasmonic acid (JA), and brassinosteroids (BR)[26,27]. The potential cross-talk between ethylene and other phytohormones could also be observed in transcriptome profiling of developing ears of qEL7SL17 and qEL7Ye478 NIL lines, at the ~2 mm stage where spikelets are being initiated. We first checked the expression of ZmACO2 homologs in two NIL lines and found another three ZmACO2 homologs expressed (FPKM > 2) in

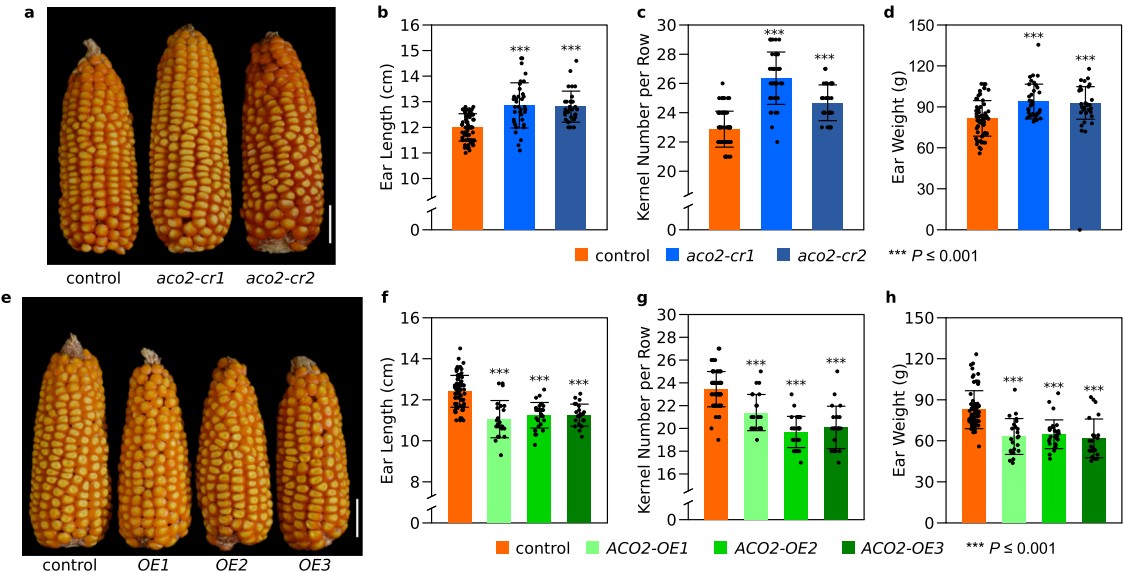

**Fig. 3 Validation of *ZmACO2* function by coding region knockout and overexpression. a** *Zmaco2* knockout alleles (*aco2-cr1* and *aco2-cr2*) made longer ears with more kernels than sibling controls; scale bar = 5 cm. Performance of control and *Zmaco2* knockout alleles (*aco2-cr1* and *aco2-cr2*) on ear length (**b**, $p = 2.37 \times 10^{-8}$ and $1.87 \times 10^{-9}$ respectively), kernel number per row (**c**, $p = 6.82 \times 10^{-20}$ and $1.44 \times 10^{-9}$ respectively) and ear weight (**d**, $p = 1.41 \times 10^{-5}$ and $1.15 \times 10^{-4}$ respectively); orange bar, control, $n = 64$ in (**b**−**d**); blue bar, *Zmaco2* alleles, $n = 36$ (*aco2-cr1*) and 31 (*aco2-cr2*) in (**b**−**d**), respectively. **e** *ZmACO2* overexpression lines made shorter ears with less kernels than sibling non-transgenic controls; scale bar = 5 cm. **f**−**h**, Performance of ear length (**f**, $p = 5.79 \times 10^{-10}$, $1.64 \times 10^{-10}$ and $2.32 \times 10^{-9}$ respectively), kernel number per row (**g**, $p = 3.76 \times 10^{-7}$, $1.99 \times 10^{-19}$ and $2.68 \times 10^{-13}$ respectively) and ear weight (**h**, $p = 5.38 \times 10^{-8}$, $1.24 \times 10^{-8}$ and $1.54 \times 10^{-8}$ respectively) in controls and *ZmACO2* overexpression lines; orange bar, control, $n = 69$ in (**f**−**h**); green bar, *ZmACO2* overexpression line, $n = 23$ (*ACO2-OE1*), 29 (*ACO2-OE2*) and 23 (*ACO2-OE3*) in (**f**−**h**), respectively. For (**b**−**d**) and (**f**−**h**), data are presented as means ± SD. ***$p$-value ≤ 0.001, from a two-tailed, two-sample t-test. $n$ is the number of ears examined in (**b**−**d**) and (**f**−**h**).

developing ears (Fig. 5a, b), suggesting a redundant role of these genes in ethylene biosynthesis. All three genes were down-regulated in $qEL7^{Ye478}$ similar to *ZmACO2* (Fig. 5a, b), and thus may contribute to the lower ethylene level in the $qEL7^{Ye478}$ long ear line.

Ethylene can stimulate auxin biosynthesis, as suggested by previous studies[26,27]. In the low ethylene $qEL7^{Ye478}$ line, the expression of seven auxin biosynthesis-related genes were significantly altered, and four of them were downregulated (Fig. 5c and Supplementary Data 6). Notably, another three known auxin-related key regulators of meristem development, *BIF4*[10], *VT2*[8] and *FASCIATED EAR4* (*FEA4*)[34], were down-regulated in $qEL7^{Ye478}$ line (Fig. 5d and Supplementary Data 6). In particular, *BIF4*, a key component of auxin hormone signaling[10], was downregulated 5.6 fold, and *VT2*, which encodes an enzyme in the tryptamine pathway for auxin biosynthesis[8] was down-regulated 1.7 fold. *FEA4*, which tends to activate genes related to auxin response[34], was downregulated 2.1 fold. We measured the IAA content in developing ears of the two NIL lines, and found a significant decrease in IAA levels in $qEL7^{Ye478}$ (Fig. 5e). Therefore, the auxin biosynthesis and signaling pathways were suppressed in the low ethylene $qEL7^{Ye478}$ line, suggesting that ethylene may modulate auxin biosynthesis in developing maize ears.

Most flowering plants produce perfect flowers containing both stamens (male organs) and pistils (female organs)[35]. Although maize florets are initially bisexual, they become unisexual male flowers in the tassel and female flowers in the ear with only stamens or pistils through the process of sex determination[5]. Previous studies suggested JA is necessary for pistil abortion and acquisition of the male characteristics of staminate spikelets[36,37]. In the long ear line $qEL7^{Ye478}$, eight JA biosynthesis-related genes were significantly altered, and seven of them were downregulated

(Fig. 5c and Supplementary Data 6). Two known JA biosynthesis genes, *Tasselseed1* (*TS1*)[36] and *Tasselseed2* (*TS2*)[37], were also downregulated 3.0 and 2.3 fold, respectively (Fig. 5d and Supplementary Data 6). We measured JA levels in developing ears of the two NIL lines. The long ear $qEL7^{Ye478}$ line had a lower level of JA, jasmonyl-isoleucine conjugate (JA-Ile), and JA precursor 12-oxophytodienoic acid (OPDA) (Fig. 5f−i), suggesting a lower JA level may block the conversion to male florets in $qEL7^{Ye478}$ ear tips. These results also suggest that ethylene might regulate JA biosynthesis in maize, in contrast to previous reports that JA predominantly acts upstream of ethylene[26,27]. Besides IAA and JA, biosynthesis-related genes of some other phyto-hormones were also significantly altered, including BR, CK, and GA (Fig. 5c and Supplementary Data 6). These results suggest that ethylene could influence the phytohormone balance in developing maize ears, however, additional studies are needed to support this hypothesis.

**Ethylene may regulate inflorescence development-related genes**. In addition to phytohormone cross-talk, 22 key genes involved in maize inflorescence development were also significantly altered in $qEL7^{Ye478}$ (Fig. 5d and Supplementary Data 6). Of them, six floral meristem determinacy regulators, such as *INDETERMI-NATE SPIKELET1* (*IDS1*) and *TASSELS REPLACE UPPER EARS1* (*TRU1*)[38,39], were downregulated 1.7-3.0 fold (Fig. 5d and Sup-plementary Data 6). Three RAMOSA-related spikelet determinacy regulators, such as *RA3* and *RA1*[40,41], were upregulated 1.6−4.9 fold (Fig. 5d and Supplementary Data 6). Besides, meristem activity regulators, such as *ZmFCP1* and *ZmCLE7*[30−32], were either down or upregulated (Fig. 5d and Supplementary Data 6). Therefore, ethylene might affect ear development by phytohormone cross-talk and regulating inflorescence development-related genes, and the underlying mechanisms need to be revealed by further studies.

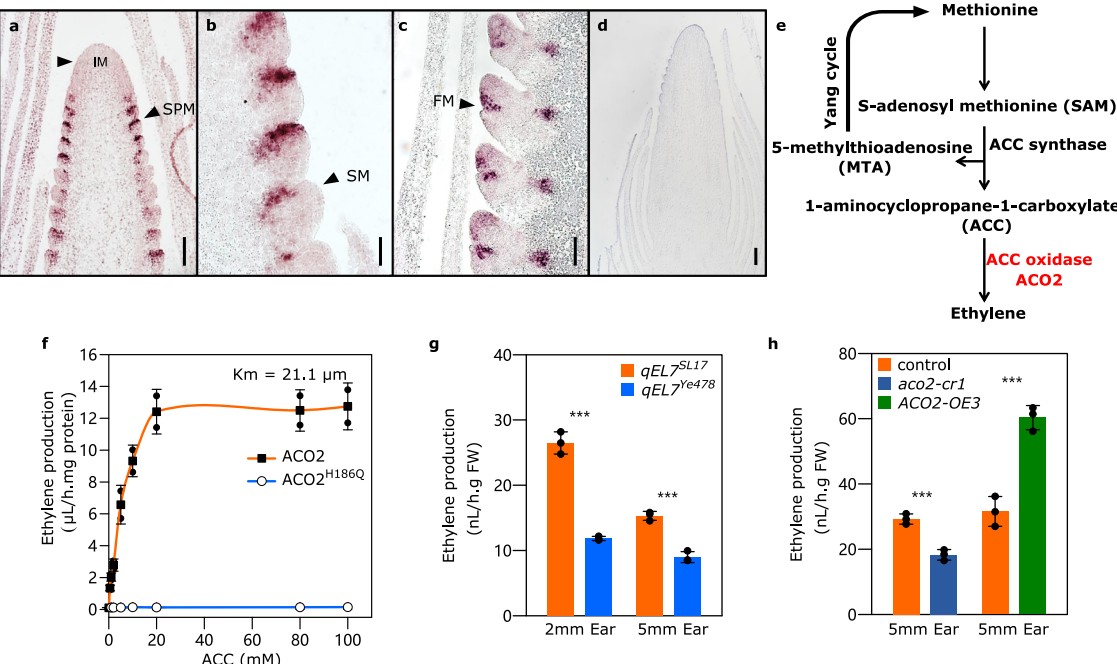

**Fig. 4 ZmACO2 is expressed in developing ears and shows enzyme activity in ethylene biosynthesis in vivo.** *ZmACO2* expression is highly localized in developing ears, revealed by RNA in situ hybridization; black arrowheads indicate SPMs (**a**), SMs (**b**), and the junction between glumes and FMs (**c**); the experiment was performed three times with similar results, using ears from at least three independent plants per experiment; negative control using sense probes (**d**); scale bars = 100 μm. **e** ACC oxidase (ACO) converts ACC to ethylene in the final pathway step of methionine-dependent ethylene biosynthesis in plants[24]. **f** Enzymatic activity of *ZmACO2* and the $H_{186}Q$ protein isoforms at various concentrations of ACC; $H_{186}Q$ represents an isoform of ZmACO2 with a substitution at amino acid 186 from a histidine (H) to a glutamine (Q); experimental data are fit to a line representing Michaelis–Menten kinetics. Error bars represent SD; three biological replicates for each. Measurement of endogenous ethylene levels in 2 and 5 mm ears of $qEL7^{SL17}$ (orange bar) and $qEL7^{Ye478}$ (blue bar) (**g**) ($p = 6.37 \times 10^{-5}$ and $2.67 \times 10^{-4}$ respectively), and 5 mm ears of *aco2-cr1* (dark blue bar) and *ACO2-OE3* (dark green bar) and their controls (orange bar) ($p = 5.50 \times 10^{-4}$ and $5.42 \times 10^{-4}$ respectively). For **g**, **h** data are presented as means ± SD. ***$p$-value ≤ 0.001, from a two-tailed, two-sample t-test, three biological replicates for each, and approximately 30 ears were used in each biological replicate.

The alteration in *ZmACO2* expression might also affect ACC content, which is the precursor of ethylene and can act directly to regulate plant growth and development[42,43]. Further evidence is needed to confirm if ACC also regulates maize ear development.

***ZmACO2 natural favorable alleles and CRISPR null alleles can enhance grain yield.*** Grain yield is a complex trait and controlled by numerous QTLs. Some of them have a considerable effect, but only in specific genetic backgrounds, which challenges their application in yield enhancement. Therefore, we evaluated the effect of *qEL7* on grain yield-related traits in diverse genetic backgrounds. We first crossed both $qEL7^{SL17}$ and $qEL7^{Ye478}$ to 13 diverse inbred lines, including eight lines carrying $Hap^{Ye478}$ and five lines carrying $Hap^{SL17}$. By comparing to the "inbred × $Hap^{SL17}$" hybrids, all 13 hybrids derived from "inbred lines × $Hap^{Ye478}$" showed longer ears with more kernels and greater ear weight (Supplementary Fig. 5). The ear weight increase ranged from 6.4 to 14.5%, with 10.8% on average (Supplementary Fig. 5). We also tested the ability of *ZmACO2* to control yield in diverse lines by crossing the *aco2-cr1* line, or its wild-type sibling, to six inbred lines; three with $Hap^{SL17}$ and three with $Hap^{Ye478}$ (Fig. 6a), and made 12 hybrids for yield measurements in three field seasons (Fig. 6b and Supplementary Data 7). We found enhancement of multiple grain yield-related traits in hybrids derived from *aco2-cr1*, including increases in ear length (3.5−7.0%), kernel number per row (4.7−6.3%), ear weight (9.0−13.9%), and grain yield per ear (5.7−21.4%) relative to hybrids derived from wild-type siblings (Fig. 6c−f and Supplementary Data 7). The average increase in grain yield per ear of 12 hybrids was about 13.4% (Fig. 6f and Supplementary Data 7).

In summary, both the natural $Hap^{Ye478}$ allele and newly created *aco2* null CRISPR allele enhance ear length, kernel number, and grain yield under diverse genetic backgrounds.

In summary, we isolated the causal gene *ZmACO2* underlying *qEL7*. We confirmed that lower *ZmACO2* expression resulted in reduced ethylene emission in vivo, but increased the number of fertile florets and kernels, possibly through cross-talk with other hormones and inflorescence development-related genes. The complete or partial loss of *ZmACO2* function through silencing the gene or modifying its promoter can enhance the grain yield of inbred lines and hybrids. Therefore, our findings provide a tool for improving grain yield by optimizing ethylene levels in maize or other cereals and expand our understanding of hormone cross-talk in inflorescence development regulation.

## Methods

**Fine mapping of qEL7**. We previously detected *qEL7* on bin7.02 using a set of $F_{2:3}$ families derived from a cross of $qEL7^{SL17} \times qEL7^{Ye478}$. Both $qEL7^{SL17}$ and $qEL7^{Ye478}$ are a set of near-isogenic lines at *qEL7* in the Ye478 genetic background[21]. To fine map *qEL7*, over 12,000 $F_2$ individuals derived from the $qEL7^{SL17} \times qEL7^{Ye478}$ were genotyped with eleven markers flanking the *qEL7* interval to identify recombinants. The heterozygous recombinant was self-crossed to segregate the homozygous recombinant (HR) and non-recombinant (HNR). The progeny families from HRs and NHRs were phenotyped at Wuhan (30°N, 114°E) and Ezhou (30°N, 114°E), China, in 2016 and 2017 under a randomized block design with three replicates. Each plot consisted of 11 individuals grown in a single row with 3 m in length, spacing of 0.3 m between plants, and 0.6 m between rows. Eight to ten competitive individuals were harvested in each plot, and subsequently air-dried to measure the ear length (cm), kernel number per row, kernel row number, ear diameter (cm) and ear weight (g). The sequences of the markers used for fine mapping were listed in Supplementary Data 8. The difference significance was examined using the two-tailed two-sample Student's t-test.

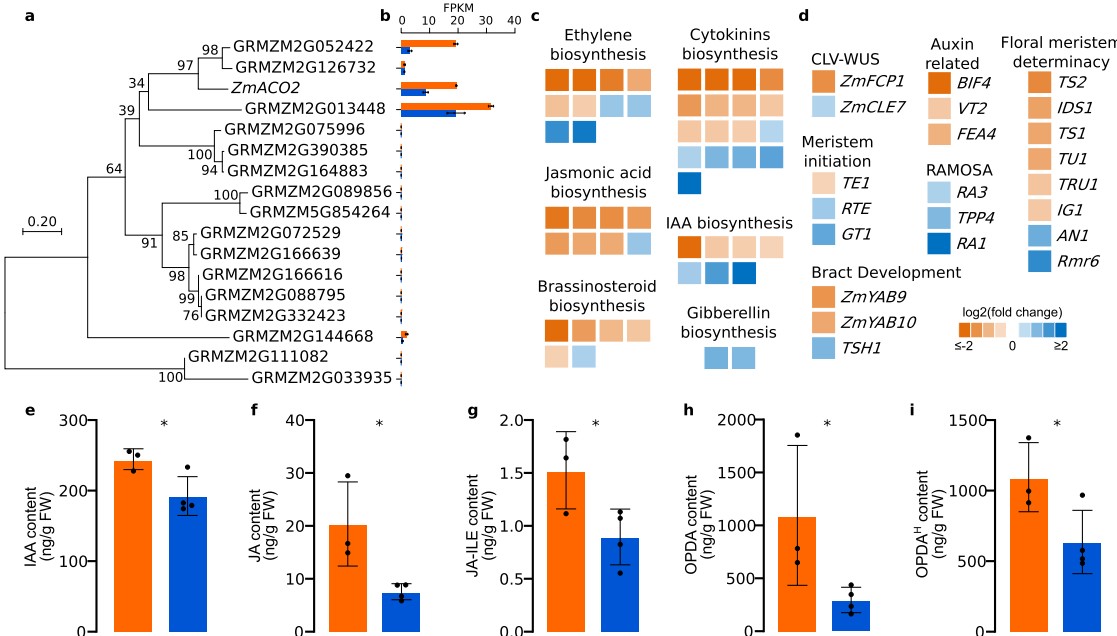

**Fig. 5 Expression profiling of developing ears in *qEL7* NIL lines revealed that ethylene might cross-talk with other phytohormones and regulate inflorescence development-related genes.** Phylogenetic tree (**a**) of homologous genes of *ZmACO2* and their expression levels (**b**) in developing ears of *qEL7* NIL lines; numbers on nodes indicate the bootstrap support of the node (1000 bootstrap replicate percentage); scale bar, branch lengths; FPKM values are shown as mean ± SD in *qEL7^SL17^* (orange bar) and *qEL7^Ye478^* (blue bar), three biological replicates for each; Expression changes of phytohormones biosynthesis-related (**c**) and inflorescence development-related (**d**) differentially expressed genes (DEGs) of in *qEL7^Ye478^* relative to *qEL7^SL17^*; the colors from orange to blue represent the $\log_2(qEL7^{Ye478}/qEL7^{SL17})$ from less than −2 to more than 2; the detailed information of gene IDs, annotation, FPKM values, *p*-values and *q*-values of these genes were listed in Supplementary Data 6. **e**−**i** Phytohormone levels in developing ears of *qEL7^SL17^* (orange bar, three biological replicates) and *qEL7^Ye478^* (blue bar, four biological replicates) including indole-3-acetic acid (IAA, **e**, *p* = 0.016), jasmonic acid (JA, **f**, *p* = 0.011), isoleucine conjugate jasmonyl-isoleucine (JA-Ile, **g**, *p* = 0.022), and JA precursor 12-oxophytodienoic acid (OPDA, **h**, **i** *p* = 0.029 and 0.025, respectively); phytohormone levels are given in ng per g of fresh weight (FW) as mean ± SD. *\*p*-value ≤ 0.05, from a two-tailed, two-sample *t*-test; approximately 30 ears were used in each biological replicate in (**e**−**i**).

**Analysis of nucleotide diversity and molecular evolution of *ZmACO2*.** A 3,539 bp region covering *ZmACO2* was resequenced in an association mapping panel with 214 diverse inbred lines by specific primers (Supplementary Data 8), including 1,230 bp upstream 1,635 bp gene body and 674 bp downstream region. Nucleotide diversity was detected by multiple sequence alignment using BioEdit version 7.2.5[44]. Those variants with minor allele frequency (MAF) over 5% were identified and were then used for an association study. All phenotypes used in this study were measured in Sanya (18°N, 109°E) in 2019 with two replicates and ~ten plants for each replicate. Association mapping was performed using a mixed linear model (MLM), considering population structure and relative kinship, in TASSEL version 3.0.67[45–47]. Pairwise linkage disequilibrium was calculated and then plotted using R software version 3.5.1[48]. A Bonferroni-corrected significance threshold ($p \leq 0.01/56 = 1.79E-04$) was used to identify the significant association. The transcription factor binding site prediction was conducted by PlantPAN 3.0[49].

The selection pressure on *ZmACO2* during maize domestication and improvement was estimated in a collection of 44 maize landraces and 53 *Z. mays* subsp. *parviglumis* teosintes (Supplementary Data 3). The *ZmACO2* genomic region was amplified and sequenced using primers listed in Supplementary Data 8. Nucleotide diversity (π) and Tajima's D were estimated using DnaSP ver. 5.0[50]. Three neutral loci (*adh1*, *adh2*, and *te1*)[51–53] were used as controls for the HKA test[54] using *Zea diploperennis* as the outgroup. The overall HKA *P*-value was obtained by summing the individual χ² values of the three control genes. All of the sequences are listed in source data.

**Vector construction, genetic transformation, and identification of the transgenic maize.** CRISPR-Cas9 was used to generate null alleles and mutations in the *ZmACO2* (Zm00001d020686) promoter region. Single-guide RNAs (sgRNAs) were designed based on the B73 reference genome sequence using the CRISPR-P web-tool[55] (http://cbi.hzau.edu.cn/crispr/). sgRNA arrays were cloned into the CPB binary vector following the manufacturer's suggested protocols[56]. To overexpress *ZmACO2*, we cloned a 1,853 bp *ZmACO2* fragment including 1,133 bp coding region and 720 bp 3′-UTR into the binary vector pZZ01523, which was driven by the maize *Ubiquitin* (Zm00001d015327) promoter. All plasmids were transformed into inbred line KN5585 via *Agrobacterium*-mediated transformation[57] at China National Seed Group Co., Ltd (Wuhan, China). Genomic editing of *ZmACO2* was screened by PCR amplification and Sanger sequencing of the target regions. The

Cas9 negative edited plants were selfed for two generations before further phenotypic scoring and expression analysis. The guide RNA sequences, PCR primers for *ZmACO2*, and its promoter genotyping are listed in Supplementary Data 8. Overexpression transgenic lines were confirmed by amplifying the target gene and *Ubiquitin* promoter with specific primers (Supplementary Data 8).

**Gene expression analysis.** *ZmACO2* expression was analyzed using developing ears that were collected from B73, two NILs, *ZmACO2* promoter-edited lines (*ACO2^CR-pro1^−ACO2^CR-pro5^*), *ZmACO2* overexpression lines (*ACO2-OE1−ACO2-OE3*), and 40 diverse inbred maize lines, respectively. Total RNA was extracted using TRIzol® Reagent (Life Technologies, Invitrogen, Carlsbad, CA, USA) according to the manufacturer's instructions. Tissue samples were collected from B73 at various developmental stages, including seedling roots, seedling internodes, immature leaves, mature leaves, 2 mm ears, 5 mm ears, and 5 mm tassels. Total RNA of *qEL7^SL17^* and *qEL7^Ye478^* was extracted from immature ear stage 1 (2 mm ears at 10-leaf stage with inflorescence meristems (IMs) and spikelet pair meristems (SPMs)), and immature ear stage 2 (5 mm ears at 12-leaf stage with IMs, SPMs, and spikelet-meristems (SMs)). Total RNA of *ZmACO2* promoter edited lines, *ZmACO2* overexpression lines, and 40 diverse maize inbred lines were extracted from immature ears at the S1 stage. DNase I (TaKaRa Biotech, Dalian, China) was used to remove genomic DNA contamination. Oligo (dT) primers and M-MLV reverse transcriptase (Invitrogen, Carlsbad, CA, USA) were used to synthesize first-strand cDNA. Quantitative real-time PCR was performed using Universal Sybr-Green Master Mix (Bio-Rad, Hercules, CA, USA) on the CFX96 Real-Time system (Bio-Rad). The maize *Actin* gene (Zm00001d010159) was used as an internal control. The relative expression of the gene was calculated by the $2^{-\Delta Ct}$ method. The primers used for quantitative real-time PCR are listed in Supplementary Data 8.

Total RNA of *qEL7^SL17^* and *qEL7^Ye478^* from immature ears was sequenced on an Illumina HiSeq 2000 system (Illumina Inc., San Diego, CA USA) in the Beijing Genomics Institute (BGI). Raw RNA-seq reads (on average ~30 million reads for each) were trimmed with Trimmomatic v.0.36[58] (parameters: ILLUMINACLIP:TruSeq3-PE. fa:2:30:10:LEADING:3 TRAILING:3 SLIDINGWINDOW:4:20 MINLEN:50) and then aligned to B73 RefGen_v3 reference using TopHat v.2.1.1[59] (parameters: --b2-sensitive --read-mismatches 2 --read-edit-dist 2 --min-anchor 8 -- splice-mismatches 0 --min-intron-length 50 --

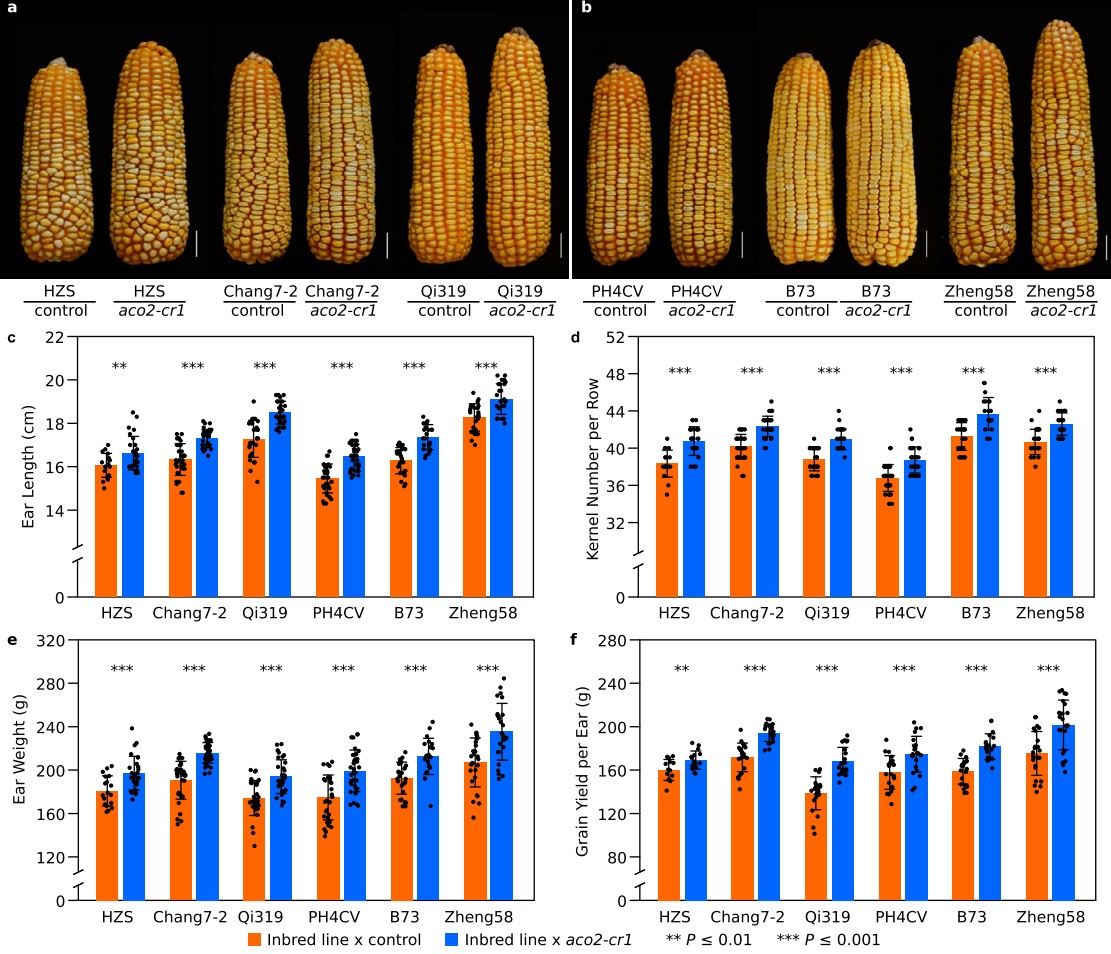

**Fig. 6 aco2-cr1 null allele enhances yield-related traits in diverse hybrid backgrounds.** Hybrids of "aco2-cr1 × inbred line" made longer ears with more kernels than "sibling control × inbred line" hybrids. Diverse inbred lines were crossed to aco2-cr1 and its sibling control; representative mature hybrid ears of aco2-cr1 and its sibling crossed with three $Hap^{SL17}$ inbred lines (**a**) and three $Hap^{Ye478}$ inbred lines (**b**); scale bar = 5 cm. **c**−**f** Performance of hybrids (**a**, **b**) in grain ear-traits including ear length (**c**, p = 0.0053, 3.14 × 10$^{-9}$, 2.15 × 10$^{-9}$, 9.50 × 10$^{-9}$, 2.50 × 10$^{-8}$ and 1.33 × 10$^{-5}$ respectively), kernel number per row (**d**, p = 1.63 × 10$^{-6}$, 1.73 × 10$^{-10}$, 5.12 × 10$^{-10}$, 4.84 × 10$^{-7}$, 1.39 × 10$^{-6}$ and 8.59 × 10$^{-7}$ respectively), ear weight (**e**, p = 3.97 × 10$^{-4}$, 1.41 × 10$^{-10}$, 2.82 × 10$^{-6}$, 3.71 × 10$^{-6}$, 2.18 × 10$^{-5}$ and 4.72 × 10$^{-5}$ respectively) and grain yield per ear (**f**, p = 0.0054, 5.97 × 10$^{-9}$, 6.83 × 10$^{-9}$, 5.01 × 10$^{-4}$, 1.43 × 10$^{-7}$ and 5.69 × 10$^{-5}$ respectively); orange bar, "sibling control × inbred line", n = 18, 34, 32, 32, 28, 27 ears, respectively; blue bar, "aco2-cr1 × inbred line", n = 28, 34, 32, 32, 22, 26 ears, respectively. Data are presented as means ± SD. **p-value ≤ 0.01, ***p-value ≤ 0.001, from a two-tailed, two-sample t-test.

max-intron-length 50,000 -- max-multihits 20). Next, Cufflinks v2.2.1[60] was used to calculate the gene expression level and call differentially expressed gene (p < 0.05, q < 0.05 and fold change > 1.5). The maize phytohormones biosynthesis-related genes were annotated by MaizeCyc v2.2[61].

Immature B73 ears (5 mm) were freshly collected and fixed in 4% paraformaldehyde (Electron Microscopy Sciences, 15714 s) for 16 h at 4 °C, and dehydrated through a graded alcohol series (50, 70, 85, 95, and 100%) and a histoclear series (National diagnostics, HS-202), then embedded in paraplast (McCormick Scientific, 39503002). 8−10 μm sections were cut using a Leica microtome (Leica RM2265), then mounted on ProbeOn Plus Slides (Fisher Scientific, 22-230-900). The probe fragment was amplified from B73 immature ears cDNA using primers ACO2-F (5′-ATGGCGCCTGCATTGTCATT-3′) and ACO2-R (5′-TCACGCGATGGCTATGCGAT-3′), and subcloned into pEASY-T1 (TransGen Biotech, CT101-01). Clones carrying an insertion in both orientations were identified and sequence-verified to generate antisense or sense probes by in vitro transcription with T7 polymerase (Sigma-Aldrich, 10881775001) and NTP cocktail. The NTP cocktail contains ribonucleoside triphosphate set (Roche, 11277057001) including ATP, GTP, CTP and 1:1 molar ratio of digoxigenin-11-uridine triphosphate (DIG-11-UTP, Roche, 11209256910): cold (unlabeled) UTP nucleotide (Roche, 11277057001). Hybridization was performed according to Jackson (1994)[62] and probes were then applied on tissue sections and incubated at 50 °C overnight. After hybridization, the sections were washed in 0.2× SSC and treated by RNaseA (Roche, 10109142001). Then the sections were incubated with anti-digoxigenin antibody (Anti-Digoxigenin-AP Fab fragments, Roche, 11093274910) at a concentration of 1:1250 for two hours at room temperature and washed in 1% Bovine Serum Albumin buffer (Sigma, B2064-50G). Sections were incubated with freshly dissolved NBT/BCIP (NBT/BCIP Ready-to-Use Tablets,

Roche, 11697471001) solution for 12−15 h before mounting, and imaged using a Nikon DS-Ri2 DIC microscope.

**In vitro ACO activity assays.** The ZmACO2 coding sequence was cloned into pET28a-sumo (Novagene), and site-directed mutagenesis was performed to introduce the point mutation $H_{186}Q$ described in the results section. Both sequences were verified by Sanger sequencing before transformation into the Rosetta E. coli strain. Cultures were grown to an $OD_{600}$ of 0.6 at 37 °C, cooled to 16 °C prior to addition of isopropyl β-D-1-thiogalactopyranoside to a final concentration of 0.5 mM, and grown for an additional 12−16 h at 16 °C. Purification of His-ZmACO2 was performed with the Ni-NTA protein purification system (QIAGEN) according to the manufacturer's instructions. Protein purity and concentration were assessed using SDS-PAGE gels.

The ZmACO2 activity assay was carried out in a scintillation vial (VWR Collection, VWRU66022-065)[24]. The standard reaction of a 5 mL mixture in a sealed 10 mL test vial contained 50 mM 4-morpholinepropanesulfonic acid (pH 7.2), 5 mM ascorbic acid sodium salt, 20 mM sodium bicarbonate, 0.02 mM iron sulfate, 1 mM ACC, 1 mM dithiothreitol, and 10% glycerol (v/v.). The reaction was initiated by adding 1 μg of purified recombinant enzyme. After incubation for 1 h at 30 °C with shaking, 1 mL of gas was withdrawn with a syringe from the headspace of the sealed 10 mL tube for ethylene quantification using an Agilent 7890B gas chromatograph fitted with a flame-ionization detector. The standard curve was created with standard concentrations of ethylene: 1, 2, 3, and 5 μL/L, with the equation $y = 12.51x - 0.914$; $r^2 = 0.998$ (y: peak area of ethylene; x: the concentration of ethylene in the sample; $r^2$: coefficient of determination). The results shown are the mean of three replicates.

**Endogenous ethylene measurement**. To measure endogenous ethylene production, we collected immature ears from the two NILs (2 and 5 mm) and transgenic lines (5 mm), respectively. Samples were placed in a 10 mL container with 2 mL LB medium and sealed with a rubber stopper. After incubation at 25 °C for 24 h in the dark, 1 mL of gas was withdrawn using a gas-tight syringe from the headspace and injected into a gas chromatograph (Agilent 7890B, USA) equipped with a flame-ionization detector and a capillary column for ethylene measurement. The ethylene production rate (microliters per gram fresh weight per hour) was calculated on the basis of the initial fresh weight of the immature ears. 30−50 plants were used for each biological replicate, and three biological replicates for each. The difference significance was calculated using a two-tailed Student's t-test.

**Measurement of endogenous IAA and JA levels**. We collected immature ears (~0.5−1 cm) from $qEL7^{SL17}$ and $qEL7^{Ye478}$ with at least three biological replicates to measure endogenous hormone levels. Each sample was weighed and ground to powder in liquid nitrogen. Approximately 100 mg powder was added to 1 mL of 1-propanol/water/concentrated HCl (2/1/0.002, v/v/v) and vortexed for 30 min at 4 °C. Then 1 mL methylene chloride ($CH_2Cl_2$) was added and vortexed for 30 min at 4 °C. After centrifugation at $13,000 \times g$ for 10 min at 4 °C, the lower layer (around 1 mL) was transferred to a new tube and dried under a stream of nitrogen. The dried samples were dissolved in 200 μL of methanol/formic acid (1/1, v/v), filtered through a 0.45 μm membrane, and transferred to sample vials for analysis in a HPLC-MS/MS system (QTRAP® 6500+ LC-MS/MS System, SCIEX) consisting of HPLC and QTRAP 6500 system. Indole-3-acetic acid (IAA) and jasmonic acid (JA), including isoleucine conjugate jasmonyl-isoleucine (JA-Ile), JA precursor 12-oxophytodienoic acid (OPDA)[63,64], were measured at National Key Laboratory of Crop Genetic Improvement, Hubei Hongshan Laboratory, Huazhong Agricultural University (Wuhan, China). The contents of IAA and JA were determined by comparison of the response to the internal standards (mol) added during extraction to the ear sample components. The hormone loss during sample preparation and chromatography was corrected by the addition of internal standards during extraction, then combined with the weight of the sample to calculate ng/g fresh weight[65].

**Evaluation of grain yield-related traits**. The phenotypes of ZmACO2 knockout lines, ZmACO2 promoter edited lines, ZmACO2 overexpression lines, and their corresponding wide-type sibling controls were investigated at Ezhou (30°N, 114°E) or Sanya (18°N, 109°E) in 2019, and Sanya (18°N, 109°E) in 2020. To evaluate alleles in different genetic backgrounds, a total of 13 lines, including 5 $Hap^{SL17}$ lines and 8 $Hap^{Ye478}$ lines, were crossed to both $qEL7^{SL17}$ and $qEL7^{Ye478}$, respectively. The 26 hybrids were grown at Ezhou (30°N, 114°E) in 2018 spring. We also separately crossed the aco2-cr1 line and its wild-type sibling to six inbred lines comprised of three $Hap^{SL17}$ lines and three $Hap^{Ye478}$ lines. The 12 hybrids were grown at Ezhou (30°N, 114°E) in 2019, Xiangyang (30°N, 112°E) and Zhangye (39°N, 100°E) in 2020. The field experiment followed a randomized block design with three replicates. Each plot consisted of 11 individuals grown in a single row of 3 m in length, spacing of 0.3 m between plants, and 0.6 m between rows. Eight to ten competitive individuals were harvested in each plot, and subsequently air-dried to measure the ear length (cm), kernel number per row and ear weight (g), and grain yield per ear (g). The difference significance was calculated using a two-tailed, two-sample Student's t-test.

**Reporting summary**. Further information on research design is available in the Nature Research Reporting Summary linked to this article.

## Data availability

Source data are available. The genetic materials that support the findings of this study and raw data used for hormone quantification are available from the corresponding authors upon request. The RNA-seq datasets are available from the National Center for Biotechnology Information, the BioProject and SRA accession numbers are [https://www.ncbi.nlm.nih.gov/bioproject/PRJNA705033] and SRR13787130 - SRR13787135. Source data are provided with this paper.

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

## Acknowledgements

This work was supported by the National Key Research and Development Program of China (2016YFD0100404), the National Natural Science Foundation of China (91935305, 31871628), the National Science Foundation (IOS-1546837 and 2129189 to D.J.), and Office of China Postdoctoral Affairs Fellowship (Fellowship to L.L.). We are grateful to Prof. Ping Yin (National Key Laboratory of Crop Genetic Improvement, Huazhong Agricultural University) for his help with protein expression and purification, and are also grateful to Xingrong Wang and Yongsheng Li (Institute of Crop Sciences, Gansu Academy of Agricultural Sciences) for their assistance in phenotype evaluation. We are also grateful to Penelope Lindsay (Cold Spring Harbor Laboratory) for help in paper editing.

## Author contributions

Z.Z., L.L. and D.J. conceived and designed the experiments. Q.N., Y.D., Y.L., X.S., H.J., R.Z., and J.Z. performed the QTL mapping. Q.N., L.L., and H.J. performed the association mapping. Q.N., Y.J., and F.Y. analyzed the gene expression. Y.J. designed the CRISPR sgRNAs and prepared the constructs for maize transformation. Q.N. performed ZmACO2 function analysis and transcriptome analysis. L.L. prepared the figures and together with Q.N. and Y.J. L.L. wrote the paper. All authors read and approved the paper.

## Competing interests

The authors declare no competing interests.
