## [Peer Review File · Nature Communications]

An ethylene biosynthesis enzyme controls quantitative variation in maize ear length and kernel yieldREVIEWER COMMENTS

Reviewer #1 (Remarks to the Author):

Increasing grain yield is the major objective in maize breeding. Ear length (EL) is highly correlated with kernel number per row, which is a key trait that determines grain yield per plant. This study reported map-based cloning of a QTL controlling EL, qEL7. The authors delimited qEL7 to a 50-kb physical interval which contains only one candidate gene ZmACO2 that encodes an ACC oxidase. Sequence and expression analysis of QTL parents and further association test in diverse inbred lines demonstrated that regulatory variations in the promoter of ZmACO2 underly qEL7. Knocking out and overexpressing ZmACO2 showed that ZmACO2 negatively regulate ear length and kernel number. Additionally, the authors also created promoter-edited alleles that exhibited decreased ZmACO2 expression and increased ear length. The authors further validated that ZmACO2 could catalyze ACC conversion into ethylene in vivo and in vitro. To examine the potential significance of qEL7/ZmACO2 on grain yield, the authors crossed qEL7 NILs and ZmACO2 knockout lines to diverse inbred lines and demonstrated that both the natural long-ear allele at qEL7 and newly created aco2 null alleles could enhance grain yield. In the end, the authors performed transcriptome profiling of qEL7 NILs and identified some possible links of cross-talk between plant hormones that balance meristem activity and floret fertility to regulate ear development. Overall, this study was well designed and performed. The findings presented in this study not only reveal novel insights into inflorescence development regulation, but also provide important targets to improve grain productivity by optimizing ethylene levels in maize or other cereals. I only have several minor questions as listed below:

- Since ZmACO2 is widely expressed in a variety of tissues, I am wondering whether qEL7 NILs and ZmACO2 knockout lines showed any other significant effects on other important agronomic traits such as plant height, flowering time, etc.
- Lines 167-186, the authors performed multiple field experiments to demonstrate that qEL7 and ZmACO2 have great potentials in enhancing maize grain yield. This part is a critical part of this study. But putting here seems to break the logic flow of this manuscript. I would suggest the authors to move this part of analysis to the end of this manuscript.
- One basic conclusion of this study is ZmACO2 regulates ear length mainly through affecting floret fertility rather than IM size. The long-ear qEL7-Ye478 NIL exhibited shorter IM, fewer florets, and higher floret fertility. To further enhance this conclusion, I would suggest the authors to add similar analyses for zmaco2 null alleles.
- To answer how ZmACO2 negatively regulates ear length, the authors performed transcriptome profiling of qEL7 NILs and found that many genes involved in auxin, JA, GA, CK and BR pathway were significantly altered. Lines 238-239, the authors stated that “a higher JA level may be causative to the male flower acquisition on ear tips of qEL7-SL17.” To enhance this speculation and clarify the cross-talk between hormones, it would be necessary to measure the level of JA (and other phytohormones if possible) in developing ears of qEL7 NILs or ZmACO2 knockout lines as the authors have done for ethylene.

- The authors previously cloned an EL QTL, qKNR6 that encodes a serine/threonine protein kinase. Are there any possible genetic or molecular relationships between qKNR6 and qEL7/ZmACO2?
- Does ZmACO2 or Hap-Ye478 show any signal of selection? How about the distribution of long-ear Hap-Ye478 in diverse maize? Is Hap-Ye478 a standing variant in teosinte or post-domestication mutation?
- Four SNPs and one 7 bp InDel in ZmACO2 promoter that are in complete LD exhibited the most significant association with EL. Are these variants located in any conserved regulatory motifs? Is there any possibility to further identify the causative variant (like transient assay)?

Reviewer #2 (Remarks to the Author):

The work of Ning et al describes the characterization of a maize QTL qEL7 that leads to a phenotype that shows a higher rate of spikelet development that gives rise to a higher kernel number (and thus yield). They identified that this QTL is linked with mutations in the promotor of ACO2, leading to a lower ACO2 expression and a reduction of ethylene production in the spikelet. They validated the role of ACO2 expression using the segregation of inbred lines, independent CRISPR knock-outs and constitutive ACO2 overexpression lines, often using field trials of several years. They also perform a hybrid cross and confirm that the interesting QTL (named Hap Ye478) leads to hybrid vigor with a higher yield (+ 13%), making the work relevant for corn breeding and production. The authors also did the essential effort to proof that ACO2 is specifically expressed in the upper spikelet meristems of the female inflorescence meristem and that ACO2 biochemically converts of ACC into ethylene. As such, it is a very nice study with excellent and robust results, with great relevance to both academia and industry.

The only drawback is that the authors did not show (or discuss) the possibility that ACC, and not ethylene, could be involved in causing this kernel phenotype. They provide evidence that the development of florets into kernels is causative of the phenotype, while a similar seed-set phenotype in Arabidopsis has been linked with ACC signaling (Tsuchisaka et al., 2009; Mou et al., 2020). My reasoning is that a lower ACO2 expression and corresponding ethylene production level, could also lead to a higher ACC content, locally in the spikelet meristem in qELYe478 or the CRISPR lines, enabling local ACC signaling in the spikelet, leading to the phenotype observed. For example, this could be tested by treating ears with 1-MCP and evaluate the role of reduced ethylene signaling (mimic the higher kernel number phenotype in qELSL17) with and without supplementing ACC. Or one could study other ethylene related mutants. Alternative experiments could be conceived to elucidate the role of ACC in this story, and be conclusive that ethylene is causal.

Below are some other suggestions.

- The section on hormonal crosstalk (p 204-236) is more speculative. The authors draw suggestive conclusions based on the number of hormonal biosynthesis genes being up or down regulated. This is dangerous, as many biosynthesis genes could involve the conjugation/degradation of active hormones and therefore their upregulation lowers the hormone level. In order to make solid conclusions about hormone levels, the authors should performed a hormonomics analysis between the qELSL17 and qELYe478, or they should revise their statements.

-Figures: instead of listing p-values (which does not mean a lot and is highly dependent on the number of replicates) above the charts, it is easier to mark significant differences (based on the 95% confidence interval) using stars or symbols in the charts. Why did the author choose this style for labeling their statistical analysis, and are they willing to reformat it to more conventional ways of presenting statistic outputs?

- Figure 1t is quite messy. The variation is super large, and the higher replicate number in HapYe478 compared to HapSL17 is giving statistical bias. It is unclear if equal variance, nor normal distribution, is achieved or tested. Based on the individual observation points presented in panel 1t, it seems unlikely. Therefore a non-parametric test is be needed instead of the two-tailed two-sampled t-test (but could lead to the same outcome).

- Figure 1u-v: the data and the r-values show that there is only a very weak correlation between ACO2 expression and ear length and kernel number per row. This data does not make the story stronger, on the other hand, it indicates that random variation in expression is equally important to be correlated to ear length and kernel number per row as a lower ACO2 expression is. I believe this data should be removed or interpreted with more caution.

- The authors should mention if other ACO genes (based on other published transcriptome data (e.g. single cell ear transcriptome data Xu et al. Developmental Cell 2021; or their own transcriptome data) are also expressed in the spikelets to rule out gene redundancy. This could help to understand why the ethylene production levels observed in the qELYe478 and CRISPR lines is not zero. Probably other ACO's are still contributing to some ethylene production in the ears. Perhaps a double (or triple) knock-out could give a more severe phenotype, leading to even more kernel yield. This should be explored or discussed.

- Supplemental Table 8 is informative, but also incomplete. It would be helpful is the authors present gene names (based on curated databases) to give the reader an idea what is the identity of the differentially expressed genes (for example, it could help to show which ethylene biosynthesis genes are up or down regulated, without me having to look up all the gene identifiers)

- Figure 4: I wonder if the authors can include data (if gathered in the field trials) about the kernel weight (100 kernel weight) to confirm that both kernel number and weight is higher in the hybrids, similar as in the near isogenic lines (Figure 1).

- We wonder if there are other phenotypes (besides the ear) in the qEL7Ye478 line, based on the fact that ACO2 is mainly expressed in mature leaves (Supl Figure 3a). There might be other leaf phenotypes such as a senescence of assimilation effect? Can the authors elaborate on this?

REVIEWER COMMENTS

Reviewer #1 (Remarks to the Author):

Increasing grain yield is the major objective in maize breeding. Ear length (EL) is highly correlated with kernel number per row, which is a key trait that determines grain yield per plant. This study reported map-based cloning of a QTL controlling EL, qEL7. The authors delimited qEL7 to a 50-kb physical interval which contains only one candidate gene ZmACO2 that encodes an ACC oxidase. Sequence and expression analysis of QTL parents and further association test in diverse inbred lines demonstrated that regulatory variations in the promoter of ZmACO2 underly qEL7. Knocking out and overexpressing ZmACO2 showed that ZmACO2 negatively regulate ear length and kernel number. Additionally, the authors also created promoter-edited alleles that exhibited decreased ZmACO2 expression and increased ear length. The authors further validated that ZmACO2 could catalyze ACC conversion into ethylene in vivo and in vitro. To examine the potential significance of qEL7/ZmACO2 on grain yield, the authors crossed qEL7 NILs and ZmACO2 knockout lines to diverse inbred lines and demonstrated that both the natural long-ear allele at qEL7 and newly created *aco2* null alleles could enhance grain yield. In the end, the authors performed transcriptome profiling of qEL7 NILs and identified some possible links of cross-talk between plant hormones that balance meristem activity and floret fertility to regulate ear development. Overall, this study was well designed and performed. The findings presented in this study not only reveal novel insights into inflorescence development regulation, but also provide important targets to improve grain productivity by optimizing ethylene levels in maize or other cereals. I only have several minor questions as listed below:

Q1: Since ZmACO2 is widely expressed in a variety of tissues, I am wondering whether qEL7 NILs and ZmACO2 knockout lines showed any other significant effects on other important agronomic traits such as plant height, flowering time, etc.

R1: We thank the reviewer for this excellent suggestion. We added measurements of some agronomic traits in *qEL7* NILs including days to heading, days to silking, days to pollen, plant height, ear leaf length and width into Supplementary Table 1 and mentioned in L65-69 as follows:

qEL7^{Ye478} plants had a significant increase in ear length, kernel number per row, 100-kernel weight and ear weight, and later flowering, compared with *qEL7*^{SL17} (Fig. 1a-g, Supplementary Table 1), although they were similar in kernel row number, ear diameter,

plant height, leaf length and width (Fig. 1d-e, Supplementary Fig. 1a, Supplementary Table 1).

Q2: Lines 167-186, the authors performed multiple field experiments to demonstrate that qEL7 and ZmACO2 have great potentials in enhancing maize grain yield. This part is a critical part of this study. But putting here seems to break the logic flow of this manuscript. I would suggest the authors to move this part of analysis to the end of this manuscript.

R2: Thanks, we have moved this part to the end of the results section of the manuscript.

Q3: One basic conclusion of this study is ZmACO2 regulates ear length mainly through affecting floret fertility rather than IM size. The long-ear qEL7-Ye478 NIL exhibited shorter IM, fewer florets, and higher floret fertility. To further enhance this conclusion, I would suggest the authors to add similar analyses for *zmaco2* null alleles.

R3: We thank the reviewer for this excellent suggestion. We did a similar analysis for the CRISPR knockout *aco2-cr1* and its wild-type sibling control, and added phenotypes of IM length and diameters, floret number per row, silky kernels per row and number of aborted florets (Supplementary Fig. 3b-g). *aco2-cr1* had a lower floret abortion rate compared to its control, consistent with the *qEL7^{Ye478}* long ear allele. However, *aco2-cr1* plants also had a larger IM size and more florets compared to the control, suggesting *aco2* knockout alleles increase kernel number by promoting both floret formation and fertility. We revised the manuscript as follows:

L130-L135

Through the analysis of developing ears, we found *aco2-cr1* plants had a lower floret abortion rate (Supplementary Fig. 3b-g), consistent with the effect of the long ear allele *qEL7^{Ye478}* in floret fertility (Supplementary Fig. 1c). However, the *aco2-cr1* plants also had a larger IM with more florets. These results suggest that *aco2* knockout alleles increase kernel number by promoting both floret formation and floret fertility.

Q4: To answer how ZmACO2 negatively regulates ear length, the authors performed transcriptome profiling of qEL7 NILs and found that many genes involved in auxin, JA, GA, CK and BR pathway were significantly altered. Lines 238-239, the authors stated that “a higher JA level may be causative to the male flower acquisition on ear tips of qEL7-SL17.” To enhance this speculation and clarify the cross-talk between

hormones, it would be necessary to measure the level of JA (and other phytohormones if possible) in developing ears of qEL7 NILs or ZmACO2 knockout lines as the authors have done for ethylene.

R4: We thank the reviewer for this excellent suggestion. We measured these hormones in developing ears of *qEL7* NIL lines, including IAA, JA, isoleucine conjugate jasmonyl-isoleucine (JA-Ile), and JA precursor 12-oxophytodienoic acid (OPDA). All of them had a lower level in *qEL7^{Ye478}*, supporting our hypothesis. We added the data into Fig. 5e-i and revised manuscript as follows:

L221-L223

We measured the IAA content in developing ears of the two NIL lines, and found a significant decrease in IAA levels in *qEL7^{Ye478}* (Fig. 5e).

L235-L245

We measured JA levels in developing ears of the two NIL lines. The long ear *qEL7^{Ye478}* line had a lower level of JA, jasmonyl-isoleucine conjugate (JA-Ile), and JA precursor 12-oxophytodienoic acid (OPDA) (Fig. 5f-i), suggesting a lower JA level may block the conversion to male florets in *qEL7^{Ye478}* ear tips. These results also suggest that ethylene might regulate JA biosynthesis in maize, in contrast to previous reports that JA predominantly acts upstream of ethylene²⁶⁻²⁷. Besides IAA and JA, biosynthesis-related genes of some other phytohormones were also significantly altered, including BR, CK and GA (Fig. 5c, Supplementary Data 6). These results suggest that ethylene could influence the phytohormone balance in developing maize ears, however, additional studies are needed to support this hypothesis.

L430-L446

Measurement of endogenous IAA and JA levels

We collected immature ears (~0.5 to 1 cm) from *qEL7^{SL17}* and *qEL7^{Ye478}* with at least three biological replicates to measure endogenous hormone levels. Each sample was weighed and ground to powder in liquid nitrogen. The powder was added to 1 mL of 1-propanol/water/concentrated HCl (2/1/0.002, v/v/v) and vortexed for 30 min at 4°C. Then 1 mL methylene chloride (CH₂Cl₂) was added and vortexed for 30 min at 4°C. After centrifugation at 13,000g for 10 min at 4°C, the lower layer (around 1 mL) was transferred to a new tube and dried under a stream of nitrogen. The dried samples were dissolved in 200 µL of methanol/formic acid (1/1, v/v), filtered through a 0.45 µm membrane and transferred to sample vials for analysis in a HPLC-MS/MS system consisting of HPLC and QTRAP 6500 system. Indole-3-acetic acid (IAA) and jasmonic acid (JA), including

isoleucine conjugate jasmonyl-isoleucine (JA-Ile), JA precursor 12-oxophytodienoic acid (OPDA)⁶³⁻⁶⁴, were measured at National Key Laboratory of Crop Genetic Improvement, Hubei Hongshan Laboratory, Huazhong Agricultural University (Wuhan, China). The content of IAA and JA was determined using the standard method and combined with the weight of the sample to calculate ng/g FW. IAA and JA quantification were performed as previously described⁶⁵.

Q5: The authors previously cloned an EL QTL, qKNR6 that encodes a serine/threonine protein kinase. Are there any possible genetic or molecular relationships between qKNR6 and qEL7/ZmACO2?

R5: We thank the reviewer for this excellent question. Our previous studies suggest that KNR6 can interact with an Arf GTPase-activating protein (AGAP), and its phosphorylation by KNR6 affects ear length and kernel number. AGAP functions in auxin localization and transport. The expression of auxin biosynthesis genes, and IAA levels were significantly different in *qEL7* NILs. So *qEL7* and *qKNR6* may have genetic or molecular relationships, linked by auxin. This could be tested by crossing the *KNR6* NILs with *qEL7* NILs and phenotyping in the F2 population. However, we don't have such data, and it would take at least one year to develop the population and collect phenotypes. Therefore, we are not able to address this question in the revision.

Q6: Does ZmACO2 or Hap-Ye478 show any signal of selection? How about the distribution of long-ear Hap-Ye478 in diverse maize? Is Hap-Ye478 a standing variant in teosinte or post-domestication mutation?

R6: We thank the reviewer for this excellent question. To examine the selection of this region during maize domestication and improvement, we sequenced the *ZmACO2* promoter and gene body in 53 teosinte and 44 maize landrace lines. The promoter region, covering the five associated sites, showed a weak selection signal with a considerable reduction in nucleotide diversity ($\pi_{\text{maize}}/\pi_{\text{teosinte}} = 0.33$) and a non-neutral evolution pattern ($p = 0.01$, HKA test) from teosinte to maize. The *Hap*^{Ye478} had already emerged before domestication, and was maintained at a high frequency during maize domestication and improvement. We added these results into Supplementary Figure 2 and mentioned these in the manuscript as follows:

L113-L123

We sequenced the *ZmACO2* promoter and gene body from 53 teosinte and 44 maize landrace lines to examine the selection pressure acting on this region during maize

domestication and improvement (Supplementary Data 3). The promoter region, covering the five associated sites, showed a weak signal of selection, with a considerable reduction in nucleotide diversity ($\pi_{\text{maize}}/\pi_{\text{teosinte}} = 0.33$) and a non-neutral evolution pattern ($p = 0.01$, HKA test) from teosinte to maize (Supplementary Fig. 2a). However, the favorable *Hap*^{Ye478} haplotype wasn't obviously enriched from teosinte to maize, as it was present in over 50% of both teosinte and maize lines (Supplementary Fig. 2b), indicating that *Hap*^{Ye478} emerged before domestication and was maintained at a high frequency during maize domestication and improvement.

L322-L330 in method section

The selection pressure on *ZmACO2* during maize domestication and improvement was estimated in a collection of 44 maize landraces and 53 *Z. mays* subsp. *parviglumis* teosintes (Supplementary Data 3). The *ZmACO2* genomic region was amplified and sequenced using primers listed in Supplementary Data 8. Nucleotide diversity (π) and Tajima's D were estimated using DnaSP ver. 5.0⁵⁰. Three neutral loci (*adh1*, *adh2* and *te1*)⁵¹⁻⁵³ were used as controls for the HKA test⁵⁴ using *Zea diploperennis* as the outgroup. The overall HKA *P*-value was obtained by summing the individual χ^2 values of the three control genes. All of the sequences have been deposited in NCBI Genbank MZ644111-MZ644981.

Q7: Four SNPs and one 7 bp InDel in *ZmACO2* promoter that are in complete LD exhibited the most significant association with EL. Are these variants located in any conserved regulatory motifs? Is there any possibility to further identify the causative variant (like transient assay)?

R7: Thanks for this question. We predicted the transcription factor binding (TF) motifs for the 7 bp InDel region and found that it could introduce new binding sites for bHLH, TCP and Dehydrin TF families. RNA *in situ* hybridization in developing ears revealed that *ZmACO2* has a highly localized expression pattern, suggesting it is induced by TF proteins with similar expression patterns. However, to examine the impact of the 7 bp InDel on *ZmACO2* promoter efficiency, for example by transient assays, we would need to conduct the assays using *ZmACO2* expressing cells or cells co-expressing TFs that bind to the 7 bp InDel region. However, these are very challenging experiments, and we don't know the TFs proteins that bind to the 7 bp InDel region, so we are not able to validate the causative variant. We have added the result of TF binding site prediction into the manuscript as follows:

L103-L105

Transcription factor (TF) binding motif predictions showed that the 7 bp insertion could introduce new binding sites for bHLH, TCP and Dehydrin TF families (Supplementary Table 3).

Reviewer #2 (Remarks to the Author):

The work of Ning et al describes the characterization of a maize QTL *qEL7* that leads to a phenotype that shows a higher rate of spikelet development that gives rise to a higher kernel number (and thus yield). They identified that this QTL is linked with mutations in the promotor of *ACO2*, leading to a lower *ACO2* expression and a reduction of ethylene production in the spikelet. They validated the role of *ACO2* expression using the segregation of inbred lines, independent CRISPR knock-outs and constitutive *ACO2* overexpression lines, often using field trials of several years. They also perform a hybrid cross and confirm that the interesting QTL (named Hap Ye478) leads to hybrid vigor with a higher yield (+ 13%), making the work relevant for corn breeding and production. The authors also did the essential effort to proof that *ACO2* is specifically expressed in the upper spikelet meristems of the female inflorescence meristem and that *ACO2* biochemically converts of ACC into ethylene. As such, it is a very nice study with excellent and robust results, with great relevance to both academia and industry.

Q8: The only drawback is that the authors did not show (or discuss) the possibility that ACC, and not ethylene, could be involved in causing this kernel phenotype. They provide evidence that the development of florets into kernels is causative of the phenotype, while a similar seed-set phenotype in Arabidopsis has been linked with ACC signaling (Tsuchisaka et al., 2009; Mou et al., 2020). My reasoning is that a lower *ACO2* expression and corresponding ethylene production level, could also lead to a higher ACC content, locally in the spikelet meristem in *qELYe478* or the CRISPR lines, enabling local ACC signaling in the spikelet, leading to the phenotype observed. For example, this could be tested by treating ears with 1-MCP and evaluate the role of reduced ethylene signaling (mimic the higher kernel number phenotype in *qELSL17*) with and without supplementing ACC. Or one could study other ethylene related mutants. Alternative experiments could be conceived to elucidate the role of ACC in this story, and be conclusive that ethylene is causal.

R8: This is a very good suggestion. We agree that our results can not determine the causal molecular for the function of *qEL7*, ACC or ethylene. However, the developing ear treatment by 1-MCP or ACC is very challenging in maize since the developing ear is tightly covered by many layers of husk and sheath leaves. There are almost no reports that treated developing maize ears with hormones. Therefore, we may not be able to conduct the experiments to address the excellent suggestion by the reviewer. We added two sentences in the manuscript to discuss the possibility that ACC, not ethylene, is casual.

L256-L260

The alteration in *ZmACO2* expression might also affect ACC content, which is the precursor of ethylene and itself directly to regulate plant growth and development⁴²⁻⁴³. Further evidence is needed to confirm if ACC also regulates maize ear development.

Below are some other suggestions.

Q9: The section on hormonal crosstalk (p 204-236) is more speculative. The authors draw suggestive conclusions based on the number of hormonal biosynthesis genes being up or down regulated. This is dangerous, as many biosynthesis genes could involve the conjugation/degradation of active hormones and therefore their upregulation lowers the hormone level. In order to make solid conclusions about hormone levels, the authors should performed a hormonomics analysis between the qELSL17 and qELYe478, or they should revise their statements.

R9: Thanks. We measured IAA, JA and biosynthetic precursors of JA in developing ears of *qEL7* NIL lines. They all showed a lower level in *qEL7*^{Ye478}, which could support our hypothesis here. However, we currently do not have enough developing ear tissue to measure the other plant hormones. So we revised the statement in the manuscript as follows:

L221-L223

We further measured the IAA content in the developing ear of two NIL lines, and the IAA content showed a significant decrease in *qEL7*^{Ye478} (Fig. 5e).

L235-L245

We measured JA levels in developing ears of the two NIL lines. The long ear *qEL7*^{Ye478} line had a lower level of JA, jasmonyl-isoleucine conjugate (JA-Ile), and JA precursor 12-oxophytodienoic acid (OPDA) (Fig. 5f-i), suggesting a lower JA level may block the conversion to male florets in *qEL7*^{Ye478} ear tips. These results also suggest that ethylene might regulate JA biosynthesis in maize, in contrast to previous reports that JA predominantly acts upstream of ethylene²⁶⁻²⁷. Besides IAA and JA, biosynthesis-related genes of some other phytohormones were also significantly altered, including BR, CK and GA (Fig. 5c, Supplementary Data 6). These results suggest that ethylene could influence the phytohormone balance in developing maize ears, however, additional studies are needed to support this hypothesis.

Q10: Figures: instead of listing p-values (which does not mean a lot and is highly dependent on the number of replicates) above the charts, it is easier to mark significant differences (based on the 95% confidence interval) using stars or symbols in the charts. Why did the author choose this style for labeling their statistical analysis, and are they willing to reformat it to more conventional ways of presenting statistic outputs?

R10: We thank the reviewer for this excellent suggestion. We changed the figures and now use asterisks to mark significant differences and list the *p*-values in the legend.

Q11: Figure 1t is quite messy. The variation is super large, and the higher replicate number in HapYe478 compared to HapSL17 is giving statistical bias. It is unclear if equal variance, nor normal distribution, is achieved or tested. Based on the individual observation points presented in panel 1t, it seems unlikely. Therefore a non-parametric test is be needed instead of the two-tailed two-sampled t-test (but could lead to the same outcome).

R11: We thank the reviewer for this good question. We conducted a Kruskal–Wallis test instead of t-test for this dataset, and it led to a similar result, that *ZmACO2* expression in lines with the *Hap*^{Ye478} was significantly lower than in the *Hap*^{SL17} lines ($H=10.26$, $p=0.0014$). We added the Kruskal–Wallis test result to the figure panel.

Q12: Figure 1u-v: the data and the r-values show that there is only a very weak correlation between ACO2 expression and ear length and kernel number per row. This data does not make the story stronger, on the other hand, it indicates that random variation in expression is equally important to be correlated to ear length and kernel number per row as a lower ACO2 expression is. I believe this data should be removed or interpreted with more caution.

R12: We thank the reviewer for this excellent suggestion. We agree, and have removed these data.

Q13: The authors should mention if other ACO genes (based on other published transcriptome data (e.g. single cell ear transcriptome data Xu et al. Developmental Cell 2021; or their own transcriptome data) are also expressed in the spikelets to rule out gene redundancy. This could help to understand why the ethylene production levels observed in the qELYe478 and CRISPR lines is not zero. Probably other ACO's are still contributing to some ethylene production in the ears. Perhaps a double (or

triple) knock-out could give a more severe phenotype, leading to even more kernel yield. This should be explored or discussed.

R13: We thank the reviewer for this excellent suggestion. We found that 16 maize genes are homologous with *ZmACO2*. We checked the expression of these genes in our RNAseq data and found three *ZmACO2* homologs expressed (FPKM > 2) in developing ears. We added these data into Fig. 5a-b and mentioned in the manuscript as follows:

L206-L211

We first checked the expression of *ZmACO2* homologs in two NIL lines and found another three *ZmACO2* homologs expressed (FPKM > 2) in developing ears (Fig.5 a-b), suggesting a redundant role of these genes in ethylene biosynthesis. All three genes were down-regulated in *qEL7^{Ye478}* similar to *ZmACO2* (Fig.5 a-b), and thus may contribute to the lower ethylene level in the *qEL7^{Ye478}* long ear line.

Q14: Supplemental Table 8 is informative, but also incomplete. It would be helpful if the authors present gene names (based on curated databases) to give the reader an idea what is the identity of the differentially expressed genes (for example, it could help to show which ethylene biosynthesis genes are up or down regulated, without me having to look up all the gene identifiers)

R14: We thank the reviewer for this good suggestion. However, the maize gene function annotation is not as good as in *Arabidopsis*, and most genes don't have a name. So we added the best hits of these genes in *Arabidopsis* and their gene names. We also added a column to show up or down-regulation of these genes in *qEL7^{Ye478}*. All the information has been added into Supplementary Data 6.

Q15: Figure 4: I wonder if the authors can include data (if gathered in the field trials) about the kernel weight (100 kernel weight) to confirm that both kernel number and weight is higher in the hybrids, similar as in the near isogenic lines (Figure 1).

R15: We thank the reviewer for this excellent suggestion. We didn't measure the 100 kernel weight for the hybrid populations, so we cannot add this data.

Q16: We wonder if there are other phenotypes (besides the ear) in the *qEL7Ye478* line, based on the fact that *ACO2* is mainly expressed in mature leaves (Supl Figure 3a). There might be other leaf phenotypes such as a senescence of assimilation effect? Can the authors elaborate on this?

R16: We thank the reviewer for this excellent suggestion. As in our response to the other reviewer, we added phenotypes of some agronomic traits in *qEL7* NILs, including days to heading, days to silking, days to pollen, plant height, ear leaf length and width into Supplementary Table 1 and mentioned in L65-69 as follows:

qEL7^{Ye478} plants had a significant increase in ear length, kernel number per row, 100-kernel weight and ear weight, and later flowering, compared with *qEL7^{SL17}* (Fig. 1a-g, Supplementary Table 1), although they were similar in kernel row number, ear diameter, plant height, leaf length and width (Fig. 1d-e, Supplementary Fig. 1a, Supplementary Table 1).

We also checked leaves of *aco2-cr1* and wild type control, and we did not observe any differences, including in senescence. We added an image of *aco2-cr1* and control leaves into Supplementary Fig. 3h and mentioned in L135-L137 as follows:

The *aco2* knockout also did not impact other leaf phenotypes, such as senescence (Supplementary Fig. 3h).

REVIEWERS' COMMENTS

Reviewer #1 (Remarks to the Author):

The authors have adequately addressed my previous questions. I donot have further questions. It's an excellent paper!

Reviewer #2 (Remarks to the Author):

The authors have nicely addressed all my main concerns and questions. I highly appreciate the extra experiments that were conducted to make the data even better (I especially appreciate the additional hormone analyses confirming the pre-stated hypothesis). The authors also took into account the statistical suggestions. Excellent rebuttal.